# Switching magnon chirality in artificial ferrimagnet

Yahui Liu[1], Zhengmeng Xu[1], Lin Liu[1], Kai Zhang[1], Yang Meng [2,3✉], Yuanwei Sun[1,4], Peng Gao [1,4], Hong-Wu Zhao [2,3,5], Qian Niu[6✉] & J. Li [1✉]

Chirality, an intrinsic degree of freedom, has been barely exploited as the information carriers in data transmission, processing, computing, etc. Recently the magnons in antiferromagnets were proposed to carry both right-handed and left-handed chiralities, shedding a light on chirality-based spintronics in which chirality-based computing architectures and chiral magnonic devices may become feasible. However, the practical platform for chirality-based spintronics remains absent yet. Here we report an artificial ferrimagnetic Py/Gd/Py/Gd/Py/Pt multilayer by which the switching, reading, and modulation of magnon chirality are demonstrated. In particular, the coexisting resonance modes of ferromagnetic and anti-ferromagnetic characteristics permit the high adjustability and easy control of magnon chirality. As a main result, we unambiguously demonstrated that Py precessions with opposite chiralities pump spin currents of opposite spin polarizations into the Pt layer. Our result manifests the chirality as an independent degree of freedom and illustrates a practical magnonic platform for exploiting chirality, paving the way for chirality-based spintronics.

[1] International Center for Quantum Materials, School of Physics, Peking University, 100871 Beijing, China. [2] Beijing National Laboratory for Condensed Matter Physics, Institute of Physics, Chinese Academy of Sciences, 100190 Beijing, China. [3] School of Physical Sciences, University of Chinese Academy of Sciences, 100049 Beijing, China. [4] Electron Microscopy Laboratory, School of Physics, Peking University, 100871 Beijing, China. [5] Songshan Lake Materials Laboratory, Dongguan 523808, China. [6] Department of Physics, University of Science and Technology of China, Hefei 230026, China. ✉email: ymeng@iphy.ac.cn; niuqian@ustc.edu.cn; jiali83@pku.edu.cn

In condensed matter physics, the chirality of elementary particles and quasiparticles plays an important role in many unconventional phenomena, such as the quantum Hall effect[1] and chiral phonon excitations[2]. In magnetic materials, magnons (quantized spin excitations) are chiral quasiparticles. While ferromagnets (FMs) only support right-handed magnon chirality, magnons in antiferromagnets (AFMs)[3–5] can support both right-handed and left-handed chiralities[6,7], which shed a light on chirality-based spintronics[8,9]. The chirality of magnons is a novel degree of freedom in spintronics, with respect to charge and spin degrees of freedom of an electron. In particular, the linear combinations of right-handed and left-handed chiralities may produce rich possible states of high-dimensionality[8] (i.e., magnonic isospin) which would facilitate the chirality-based computing[8,10], field-effect transistor[11], logic devices[12], etc. Meanwhile, the magnon chirality is a more robust degree of freedom in contrast to the amplitude and phase. The constant refreshing of the signal power is unnecessary in chirality-based spintronic devices, contrary to FM magnonic devices[8]. Therefore, chiral magnons are the prospective information carriers in spintronics with low-energy cost and high-dimensionality. However, given the pervasive incoherent effects in natural AFMs[13], as well as the current bottlenecks for exploiting the magnon chirality as an independent degree of freedom, it is urgent to find a practical platform for the chirality-based spintronics where the magnon chirality could be readily controlled and definitely measured. Ferrimagnets (FiMs), which combine the characteristics of FM and AFM, offer a new opportunity for exploiting the magnon chirality. The compensation phenomena of FiM materials, especially the dynamical aspects such as the complex spin Seebeck effect[14], the enhancement of domain wall mobility[15], the gyromagnetic reversal[16], have been reported in recent years.

In this work, we demonstrate the switching, reading, and modulation of magnon chirality in an artificial FiM, which permits the high adjustability and easy control of magnon chirality. In particular, the chirality-dependent spin pumping of FM and AFM characteristic resonance modes are discovered in coexistence, where the polarity of spin pumping is determined by the chirality of Py magnetization precession rather than Py magnetization equilibrium direction (Fig. 1a, b). This result manifests the magnon chirality as an independent degree of freedom from the magnetization equilibrium direction, i.e., the information can be carried by the chirality independently, which is vital for chirality-based spintronics. Our discovery illustrates a flexible magnonic platform that may unleash the full potential of chiral magnons in chirality-based spintronics.

## Results

**Design of artificial FiMs.** An artificial FiM consists of two magnetic sublayers, $\mathbf{M}_A$ and $\mathbf{M}_B$, with AFM coupling. When these two magnetic sublayers are uncompensated and in perfectly antiparallel alignment, the net moment ($|\mathbf{M}_A + \mathbf{M}_B|$) behaves as a FM. In ferromagnetic resonance (FMR), the net moment precesses around an external magnetic field $H$ with right-handed chirality, leading to right-handed chirality of the greater moment (master) between $\mathbf{M}_A$ and $\mathbf{M}_B$ and left-handed chirality of the weaker moment (slave) around their equilibrium directions. We can reach the $\mathbf{M}_A$ master ($|\mathbf{M}_A| > |\mathbf{M}_B|$) or $\mathbf{M}_B$ master ($|\mathbf{M}_A| < |\mathbf{M}_B|$) phase by resizing $\mathbf{M}_A$ and $\mathbf{M}_B$ in the artificial FiM, resulting in the right-handed or left-handed chirality of $\mathbf{M}_A$ precession about the $+\mathbf{z}$ direction (Fig. 1c, d). This precession mode will be referred to as the FMR mode. In the $\mathbf{M}_B$ master phase, a sufficiently strong magnetic field may twist $\mathbf{M}_A$ and $\mathbf{M}_B$ into a twisted state. When driving $\mathbf{M}_A$ and $\mathbf{M}_B$ into resonance, the AFM coupling of $\mathbf{M}_A$ and $\mathbf{M}_B$ leads to the right-handed chirality of $\mathbf{M}_A$ precession about the $+\mathbf{z}$ direction (Fig. 1e), which will be referred to as the exchange mode[17,18] in the following discussion. This exchange mode is the resonance mode of AFM character, in comparison to the FMR mode of FM character. Hence, the chirality of $\mathbf{M}_A$ precession can be manipulated by resizing or twisting $\mathbf{M}_A$ and $\mathbf{M}_B$. Note that, as shown in Fig. 1f, $\mathbf{M}_A$ and $\mathbf{M}_B$ equilibrium directions are not reversed during the switching of the chirality, illustrating the chirality as an independent degree of freedom from the magnetization equilibrium direction.

To probe spin pumping solely from $\mathbf{M}_A$ or $\mathbf{M}_B$, we synthesized an artificial FiM by spatially separating $\mathbf{M}_A$ and $\mathbf{M}_B$ into antiferromagnetically coupled $\mathbf{M}_A/\mathbf{M}_B$ multilayers so that spin pumping from the $\mathbf{M}_A$ layer could be selected explicitly by growing the spin-current receiving layer (HM) next to the $\mathbf{M}_A$ layer. Then, the spin pumping of the right-handed or left-handed

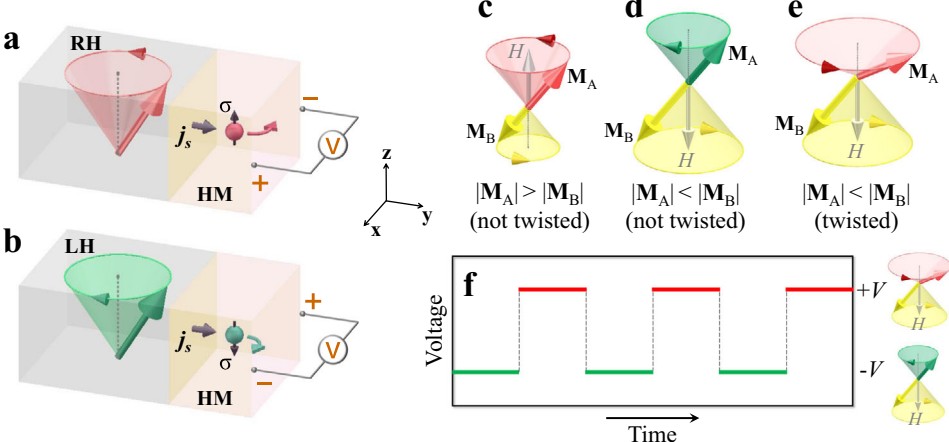

**Fig. 1 Schematic of chirality-dependent spin pumping.** Magnetization precessions with (**a**) right-handed (RH) and (**b**) left-handed (LH) chiralities around the equilibrium axis ($+\mathbf{z}$ axis) pump spin current into a heavy metal (HM) layer. The chirality is defined by the precessing magnetization and the equilibrium direction. The DC voltage polarity detects the spin polarization ($+\mathbf{z}$ or $-\mathbf{z}$) of the spin current. Here only sketches the Py magnetization precession of the outermost Py layer in Py/Gd multilayer. **c–e** sketch the magnetization precessions of two magnetic sublayers $\mathbf{M}_A$ and $\mathbf{M}_B$ with AFM coupling. When $\mathbf{M}_A$ and $\mathbf{M}_B$ are collinearly aligned, as shown in **c**, **d**, magnon chirality can be switched between RH and LH by resizing $\mathbf{M}_A$ and $\mathbf{M}_B$. LH chirality of $\mathbf{M}_A$ precession in **d** can be switched to RH chirality in **e** by twisting $\mathbf{M}_A$ and $\mathbf{M}_B$ into a twisted state. **f** By twisting $\mathbf{M}_A$ and $\mathbf{M}_B$ recurrently, the magnon chirality can be modulated and read out in form of the spin pumping voltage.

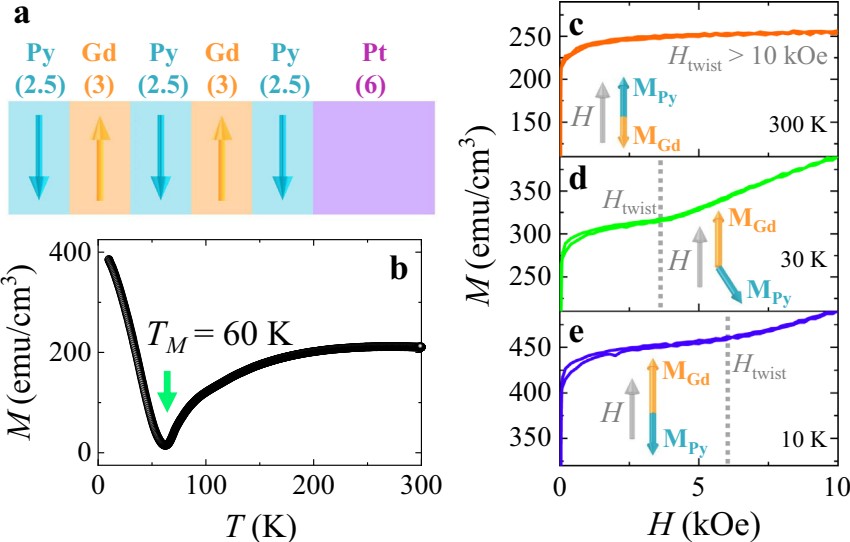

**Fig. 2 Static magnetization of Py/Gd multilayer. a** Sketch of the Py/Gd/Py/Gd/Py/Pt sample (the numbers in parentheses are thicknesses in units of nanometers). **b** Temperature dependence of in-plane magnetization at $H = 50$ Oe; the green arrow indicates the compensation temperature $T_M$. The positive half branches of the hysteresis loops at (**c**) $T = 300$ K, (**d**) $T = 30$ K and (**e**) $T = 10$ K. The twisted state is achieved at $H > H_{twist}$ (gray dotted lines in **d**, **e**), while the Gd-aligned phase is retained at $H < H_{twist}$ at $T = 30$ K and 10 K. At $T = 300$ K, only the Py-aligned phase is achievable within the $H$ range in our experiments. The Py-aligned phase, twisted state and Gd-aligned phase are illustrated in **c**–**e**. The perfect antiparallel alignment of $\mathbf{M}_{Py}$ and $\mathbf{M}_{Gd}$ is ensured in the Py-aligned phase and Gd-aligned phase.

$\mathbf{M}_A$ precession can be specified in a quantitative manner[19,20] (Fig. 1a, b). Meanwhile, the resonance frequency of artificial FiMs could be much lower than that of natural AFMs, facilitating the research of high-frequency spintronics. The aforementioned merits are unattainable in natural AFMs, making artificial FiMs the ideal platform to model genuine AFM/FiM systems for chirality-based spintronics.

**Static magnetization of the artificial FiM (Py/Gd multilayer).** Figure 2a illustrates Py(2.5)/Gd(3)/Py(2.5)/Gd(3)/Py(2.5)/Pt(6) multilayer (in nm), hereafter simplified as the Py/Gd multilayer. Different Curie temperatures of Py and Gd (for bulk materials, $T_C^{Py} = 872$ K and $T_C^{Gd} = 293$ K) accompanied by strong interfacial AFM coupling result in a compensation temperature $T_M$[21]. At $T = T_M$, the magnetic moments of Py ($\mathbf{M}_{Py}$) and Gd ($\mathbf{M}_{Gd}$) are fully compensated. For $T > T_M$, we could achieve a Py-aligned phase ($|\mathbf{M}_{Py}| > |\mathbf{M}_{Gd}|$) with $\mathbf{M}_{Py}$ parallel to $H$. For $T < T_M$, the Gd-aligned phase ($|\mathbf{M}_{Py}| < |\mathbf{M}_{Gd}|$) with $\mathbf{M}_{Py}$ opposite to $\mathbf{H}$ is accessible. With respect to the strong AFM coupling at the Py/Gd interface, the relatively weak ferromagnetic exchange in the Gd layer will result in a transition to a canted magnetic state (twisted state) in the presence of a sufficiently strong magnetic field[22]. Thus, a rich magnetic phase diagram can be achieved depending on $H$ and temperature[23,24].

Figure 2b depicts the temperature dependence of the in-plane magnetization at 50 Oe with a local minimum at $T = 60$ K, revealing that $T_M = 60$ K for the Py/Gd multilayer. Figure 2c–e exhibit the positive half branches of the hysteresis loops at $T = 300$ K, 30 K and 10 K, respectively. A nonlinear rise in the magnetization with $H$ was observed at $T = 30$ K and 10 K, indicating the initiation of the twisted state at the critical field $H_{twist}$ (gray dotted lines in Fig. 2d, e)[24]. The Gd-aligned phase is retained when $H < H_{twist}$, and the twisted state could be achieved when $H > H_{twist}$ at $T = 30$ K and 10 K. In contrast, $H_{twist}$ exceeds 10,000 Oe at $T = 300$ K; thus only the Py-aligned phase is achievable within the $H$ range at $T = 300$ K. Additionally, the Py-aligned and Gd-aligned phases can also be revealed via the measurements of anomalous Hall effect (AHE, Supplementary

Note 3). Taking $\mathbf{M}_{Py}$ and $\mathbf{M}_{Gd}$ as $\mathbf{M}_A$ and $\mathbf{M}_B$ in Fig. 1, the chirality of $\mathbf{M}_{Py}$ precession is controllable via the switching between Py-aligned phase and Gd-aligned phase, as well as the twisting of $\mathbf{M}_{Py}$ and $\mathbf{M}_{Gd}$ into the twisted state.

**Chirality-dependent spin pumping in FMR mode.** Next, we intend to manipulate the chirality of $\mathbf{M}_{Py}$ precession in FMR mode via the switching between Py-aligned phase ($T = 300$ K) and Gd-aligned phase ($T = 10$ K), and conduct spin pumping measurements. The experimental geometry is expressed in the azimuthal directions of in-plane $H$ ($\theta_H$) and $\mathbf{M}_{Py}$ equilibrium ($\theta_{Py}$). To keep $\mathbf{M}_{Py}$ along the $+\mathbf{z}$ direction ($\theta_{Py} = 0°$), it requires $\theta_H = 0°$ at $T = 300$ K and $\theta_H = 180°$ at $T = 10$ K (Fig. 3b, c). At $T = 300$ K, the master $\mathbf{M}_{Py}$ causes right-handed $\mathbf{M}_{Py}$ precession about the $+\mathbf{z}$ direction (inset in Fig. 3h). At $T = 10$ K, the right-handed $\mathbf{M}_{Gd}$ precession about the $-\mathbf{z}$ direction forces $\mathbf{M}_{Py}$ to precess with left-handed chirality about the $+\mathbf{z}$ direction (inset in Fig. 3i). Thus, we accomplish the right-handed $\mathbf{M}_{Py}$ precession at $T = 300$ K and the left-handed $\mathbf{M}_{Py}$ precession at $T = 10$ K with the same $\mathbf{M}_{Py}$ equilibrium direction along the $+\mathbf{z}$ direction.

When spin pumping occurs in the Py/Gd multilayer, only the outermost Py layer (next to Pt) gives rise to the spin mixing conductance[25] at the Py/Pt interface. The inner Gd and Py layers are physically separated from the Pt layer and make null contributions to the spin mixing conductance at the Py/Pt interface, in light of the spin current penetration depth (~1 nm) in Py[26,27] and the negligible contributions of Gd[17,28]. This fact is further evidenced by a negligible spin mixing conductance at the Gd/Pt interface of control samples (Supplementary Note 8). Hence, we can probe the spin pumping voltage $V(H)$ of the specific magnetic sublayer (the outermost $\mathbf{M}_{Py}$), with either right-handed or left-handed chirality of $\mathbf{M}_{Py}$ precession.

Figure 3d, e plot the $V(H)$ signals at $T = 300$ K and $T = 10$ K, respectively. A negative $V(H)$ is observed at $T = 10$ K with respect to the positive $V(H)$ at $T = 300$ K. At both temperatures, the dispersion relations between frequency $f$ and resonance field $H_{res}$ are effectively described by the Kittel equation[29]. Thus, we confirm that $V(H)$ originates from the FMR mode of the Py/Gd

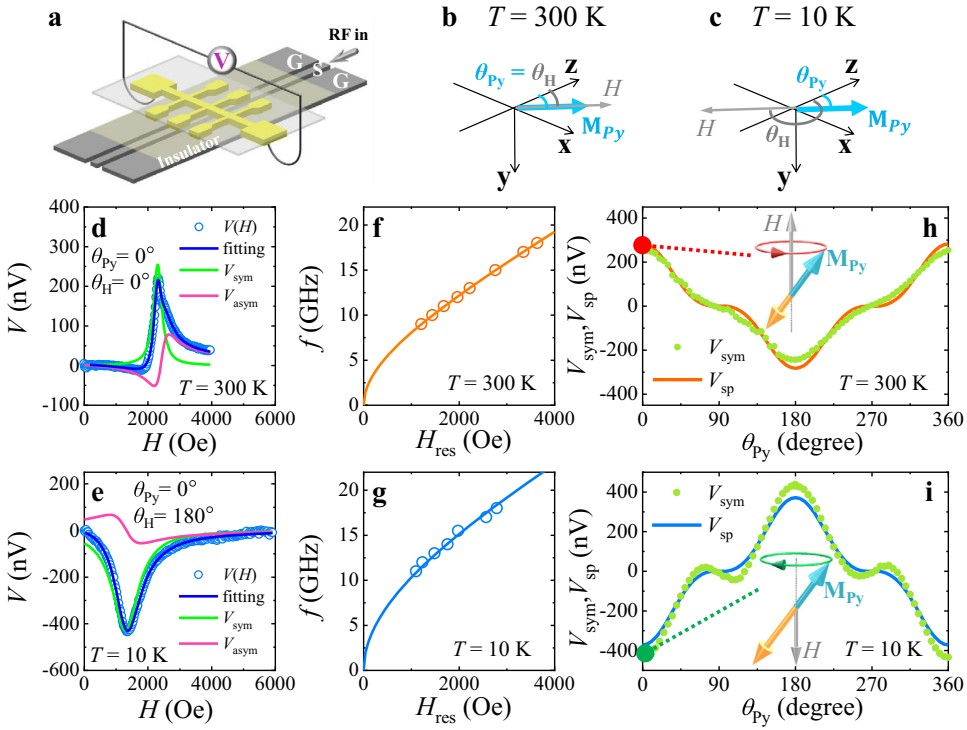

**Fig. 3 Chirality-dependent spin pumping versus spin rectification effect in the FMR mode. a** Illustration of spin pumping measurements. The experimental geometry is characterized by the azimuthal directions of $H$ ($\theta_H$) and $\mathbf{M}_{Py}$ equilibrium ($\theta_{Py}$). **b** $\mathbf{M}_{Py}$ is parallel to $H$ at $T = 300$ K. **c** $\mathbf{M}_{Py}$ is antiparallel to $H$ at $T = 10$ K. $V(H)$ signals ($f = 13$ GHz) are plotted with the fitting curves for **d** $\theta_{Py} = 0°$ and $\theta_H = 0°$ at $T = 300$ K and **e** $\theta_{Py} = 0°$ and $\theta_H = 180°$ at $T = 10$ K. $V_{asym}$ of opposite polarities (magenta lines in **d, e**) are ascribed to the spin rectification effect. The $H_{res}$ dependences of frequency $f$ (**f**) at $T = 300$ K and (**g**) at $T = 10$ K are effectively described by the Kittel equation, confirming the perfectly antiparallel alignment of $\mathbf{M}_{Py}$ and $\mathbf{M}_{Gd}$ in FMR. $\theta_{Py}$-dependent $V_{sym}$ and $V_{sp}$ are plotted at (**h**) $T = 300$ K and (**i**) $T = 10$ K. A positive $V_{sp}$ was produced by the right-handed $\mathbf{M}_{Py}$ precessions ($\theta_{Py} = 0°$ and $\theta_H = 0°$, marked by the red dot in **h**). The negative $V_{sp}$ is produced by the left-handed $\mathbf{M}_{Py}$ precession ($\theta_{Py} = 0°$ and $\theta_H = 180°$, marked by the green dot in **i**).

multilayer[29]. $\mathbf{M}_{Py}$ and $\mathbf{M}_{Gd}$ are perfectly antiparallel to each other during the precessions, leading to the right-handed $\mathbf{M}_{Py}$ precession at $T = 300$ K and the left-handed $\mathbf{M}_{Py}$ precession at $T = 10$ K. The right-handed (left-handed) $\mathbf{M}_{Py}$ precession produces the positive (negative) $V(H)$. Note that the microwave field could induce an AC current inside the FM metal. Meanwhile, the oscillating magnetization of the FM metal can induce a time dependent resistance by anisotropic magnetoresistance (AMR), AHE, etc. The interplay between the AC current and time dependent resistance may induce the spin rectification effect (SRE), leading to the DC voltage signals of symmetric and antisymmetric Lorentzians[30,31]. To exclude the contributions of SRE and quantitatively determine the spin pumping voltage $V_{sp}$ due to pure spin current, we performed angular-dependent measurements of $V(H)$. Each $V(H)$ signal is fitted by the combination of symmetric Lorentzian $V_{sym}$ and antisymmetric Lorentzian $V_{asym}$ (Fig. 3d, e). The quantitative fitting of the $\theta_{Py}$-dependent $V_{sym}$ provides the $\theta_{Py}$-dependent $V_{sp}$ (Fig. 3h, i). The details of the angular-dependent measurement and fitting are presented in Supplementary Notes 4 and 5. The tiny deviation between $\theta_{Py}$-dependent $V_{sym}$ and $\theta_{Py}$-dependent $V_{sp}$ reveals that SRE is small and $V(H)$ is mainly attributed to $V_{sp}$ (Supplementary Note 4). As plotted in Fig. 3h, i, $V_{sp}$ shows positive polarity for right-handed chirality at $T = 300$ K (red dot at $\theta_{Py} = 0°$ in Fig. 3h) in comparison to the negative polarity for left-handed chirality at $T = 10$ K (green dot at $\theta_{Py} = 0°$ in Fig. 3i). This result unambiguously demonstrates that the spin polarization of the spin current is determined by the chirality of the spin precession rather than the spin equilibrium direction. $V_{sp}$ is a good measure of chirality for spin precession.

Spin Hall angle of Pt retains the same sign in the temperature range of our measurements and makes no impact on our conclusion[32,33]. After divided by the microwave power $P_{app}$, the normalized $V_{sp}$ signal is linearly proportional to $f$ (Supplementary Note 4), in accordance with the spin pumping theory[25,34]. Hence, the pure spin current from coherent spin pumping is confirmed to be the origin of $V_{sp}$. Thermal voltage (incoherent pumping) is not evidenced in our experiments [35].

Note that, $V_{asym}$ due to SRE reverses the sign when switching the chirality of $\mathbf{M}_{Py}$ precession (magenta lines in Fig. 3d, e). According to the quantitative measurements (Supplementary Note 4), $V_{asym}$ is mainly attributed to AMR. In the scenario of AMR-related SRE[20,36,37], the radio frequency ($rf$) current $I(t) = I_{rf}\cos(2\pi f \bullet t)$ and the oscillating resistance $R(t) = R_0 - \triangle R_{AMR}\sin^2[\theta_H + \triangle\theta(t)]$ are taken into account, where $\triangle R_{AMR}$ is the magnitude of AMR and $\triangle\theta(t)$ is the time dependent cone angle of magnetization precession. The product of $I(t)$ and $R(t)$ causes a DC $V_{asym}$ which is proportional to $\cos(2\theta_H)\cos(\theta_H)\sin\Phi$, where $\Phi$ is the relative phase between $I(t)$ and $\triangle\theta(t)$[30]. The opposite chirality of magnetization precession provides the opposite $\Phi$, leading to the opposite sign of $V_{asym}$. Hence, $V_{asym}$ of SRE is also a good measure of chirality. To the best of our knowledge, it is the first time to propose a measure of magnon chirality via SRE.

**Switching the magnon chirality.** Subsequently, we intend to manipulate the chirality of $\mathbf{M}_{Py}$ precession via the twisting of $\mathbf{M}_{Py}$ and $\mathbf{M}_{Gd}$. Spin pumping measurements were performed at $T = 30$ K in the same experimental geometry of $T = 10$ K

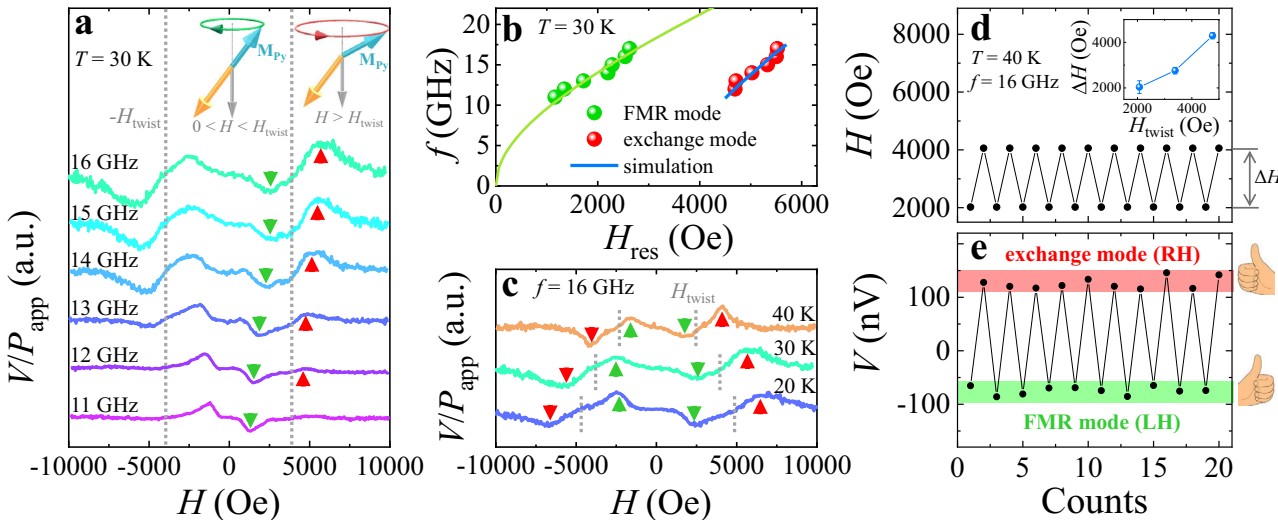

**Fig. 4 FMR mode with left-handed (LH) chirality and exchange mode with right-handed (RH) chirality. a** Normalized $V/P_{app}$ signals at $T = 30$ K. The FMR mode is observed when $H < H_{twist}$, and the exchange mode is observed when $H > H_{twist}$. **b** The dispersion relations between $f$ and $H_{res}$ of the FMR mode and exchange mode at $T = 30$ K. The FMR mode is confirmed through fitting by the Kittel equation. The $H_{res}$ dependence of $f$ for the exchange mode is calculated by the micromagnetic simulation. **c** Normalized $V/P_{app}$ signals ($f = 16$ GHz) at 20 K, 30 K and 40 K. FMR mode (exchange mode) is indicated by green (red) arrows. Gray dotted lines indicate the strength of $H_{twist}$. **d** The recurrent switching of the external field $H$ can modulate (**e**) the chirality as well as the $V(H)$ signal. The change of $H$ ($\Delta H$) for this modulation can be reduced by decreasing $H_{twist}$, as shown in the inset of **d**.

($\theta_{Py} = 0°$ and $\theta_H = 180°$). As shown in Fig. 4a, negative $V(H)$ is observed when $H < H_{twist}$ (green arrows), which corresponds to the left-handed $\mathbf{M}_{Py}$ precession of the FMR mode (confirmed via fitting by the Kittel equation in Fig. 4b. A positive $V(H)$ (red arrows) emerges in the twisted state ($H > H_{twist}$) when $f$ exceeds 12 GHz, corresponding to the second resonance mode[24]. This resonance mode is the exchange mode arising from the twisting of $\mathbf{M}_{Py}$ and $\mathbf{M}_{Gd}$ in the twisted state. The positive $V(H)$ of this mode indicates the right-handed $\mathbf{M}_{Py}$ precession, which is confirmed via micromagnetic simulation[38] (Supplementary Note 6). The coexistence of the FMR and exchange modes with opposite chiralities was observed at a series of temperatures (Fig. 4c). As $H_{twist}$ (gray dotted lines in Fig. 4c) and $H_{res}$ of the exchange mode (red arrow in Fig. 4c) decline simultaneously with increasing temperature, the correlation between the exchange mode and twisted state is unambiguously clarified (inset in Fig. 4d). It has been verified that the $V(H)$ of the FMR mode is mainly attributed to $V_{sp}$. We also examine the SRE signals of the exchange mode, and $V(H)$ of the exchange mode is mainly attributed to $V_{sp}$ (Supplementary Note 5). Overall, the left-handed $\mathbf{M}_{Py}$ precession of the FMR mode produces a negative $V_{sp}$, and the right-handed $\mathbf{M}_{Py}$ precession of the exchange mode produces a positive $V_{sp}$.

In AFMs with two magnetic sublattices $\mathbf{m}_1$ and $\mathbf{m}_2$, as specified in ref. [6], two degenerate eigenmodes at resonance are characterized by the opposite chiralities of $\mathbf{n}$ precession as well as the opposite $\langle\mathbf{m}\rangle$. Here $\mathbf{n} = (\mathbf{m}_1 - \mathbf{m}_2)/2$ is the Néel order. $\langle\mathbf{m}\rangle$ is the DC component of the precessing magnetization $\mathbf{m}$ where $\mathbf{m} = (\mathbf{m}_1 + \mathbf{m}_2)/2$ (sketched in the Fig. 1 of ref. [6]). The reversal of the $\mathbf{n}$-precession chirality is always accompanied by the reversal of $\langle\mathbf{m}\rangle$. Therefore, the chirality is not an independent degree of freedom from $\langle\mathbf{m}\rangle$. Logically speaking, the magnetic contributions of the chirality and $\langle\mathbf{m}\rangle$ are indistinguishable in such AFM systems. In contrast, Fig. 4 shows that the opposite chiralities of $\mathbf{M}_{Py}$ precession give rise to the opposite polarities of $V_{sp}$ when the $\mathbf{M}_{Py}$ and $\mathbf{M}_{Gd}$ equilibrium directions are preserved without reversals. This result unambiguously demonstrates that the chirality is an independent degree of freedom from the magnetization

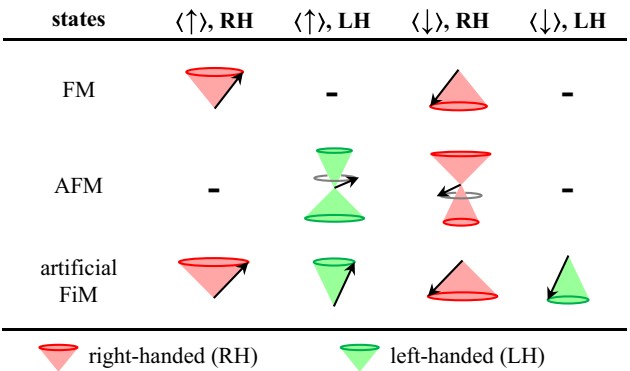

**Fig. 5 The possible states produced by the magnons in FM, AFM, artificial FiM systems.** $\langle\mathbf{m}\rangle$, $\langle\mathbf{M}\rangle$ and magnon chirality are taken into account as the degrees of freedom. The first degree of freedom is $\langle\mathbf{m}\rangle$ in AFM and $\langle\mathbf{M}\rangle$ in FM or artificial FiM. In analogy to $\langle\mathbf{m}\rangle$ in AFM, $\langle\mathbf{M}\rangle$ is the DC component of the precessing magnetization in FM or in the specific magnetic sublayer of artificial FiM (black arrows). This degree of freedom is denoted by $\langle\uparrow\rangle$ for $\langle\mathbf{m}\rangle$ // $+\mathbf{z}$ or $\langle\mathbf{M}\rangle$ // $+\mathbf{z}$, and by $\langle\downarrow\rangle$ for $\langle\mathbf{m}\rangle$ // $-\mathbf{z}$ or $\langle\mathbf{M}\rangle$ // $-\mathbf{z}$. The colorful cones denote the precessing angles of magnetic moment in FM and magnetic sublattices in AFM. In our artificial FiM, $V_{sp}$ is produced by one specific magnetic sublayer which is denoted by a single cone. Both sublattices in AFMs could have magnetic contributions so that being denoted by the double cones. The second degree of freedom is the magnon chirality which is denoted in red (RH) and green (LH). Overall, the combination of $\langle\mathbf{M}\rangle$ and magnon chirality can produce four possible states in our artificial FiM while AFMs only host two states of opposite chiralities locked with the opposite $\langle\mathbf{m}\rangle$.

direction in our artificial FiM. Namely, the information can be carried by the chirality independently. Note that, $V_{sp}$ is produced by one specific magnetic sublayer in the artificial FiM, this system can be equivalent to a FM with both right-handed and left-handed chiralities in spintronics (Fig. 5). In particular,

the combination of magnetization direction and magnon chirality in the artificial FiM can produce two more possible states than that in AFMs (Fig. 5).

To demonstrate the unique merit of the artificial FiM, we modulate the magnon chirality and monitor $V_{sp}$ concurrently. The coexistence of the FMR mode (FM characteristic) and exchange mode (AFM characteristic) provides the opportunity to modulate the chirality by tuning $H$ at the fixed frequency and temperature. Figure 4d, e show the modulation of chirality by tuning $H$ recurrently between 4 kOe and 2 kOe at $T = 40$ K and $f = 16$ GHz. The right-handed chirality (exchange mode) and left-handed chirality (FMR mode) were recurrently triggered and electrically read by measuring $V(H)$ signals, i.e., information encoded in form of the chirality can be modulated and read. The separation between two modes ($\Delta H$) can be further reduced by decreasing $H_{twist}$ so that a smaller change in $H$ would be needed (inset in Fig. 4d). This discovery promises artificial FiMs a great advantage over natural AFMs due to the ease of manipulating the magnon chirality. The coexistence of the FMR and exchange modes is also expected when $T > T_M$, which was realized in the Fe/Gd multilayer even at room temperature (an ideal working temperature for spintronics devices). We presented the detailed result in Supplementary Note 7.

In summary, chirality-dependent spin pumping was revealed in an artificial FiM. The modulation and electrical readout of the chiralities were demonstrated, which was accessible even at room temperature. Our result opens the door for the prospective applications of chiral magnons in chirality-based spintronics.

## Methods

**Sample fabrications**. The Py/Gd multilayer samples were deposited on single crystalline $Al_2O_3(0001)$ substrates by DC magnetron sputtering under an Ar pressure of 3.5 mTorr at room temperature. To obtain alternate layers of Py and Gd with different thicknesses, high purity Py (99.95%) and Gd (99.9%) targets were sputtered for different durations in sequence. The deposition rates were 2.4 nm/min and 1.2 nm/min for Gd and Py, respectively. The Pt capping layer or Cu capping layer (both 6 nm) was deposited on top of the sample to protect it from oxidation.

The Fe/Gd multilayer samples were prepared in an ultrahigh vacuum chamber with a base pressure of $2 \times 10^{-10}$ Torr. MgO substrate was annealed at 600 °C for 1 h. The Fe/Gd/Fe/Gd/Pd multilayer sample was deposited on the MgO substrate by Fe, Gd and Pd effusion cells in sequence. The sample was patterned into a Hall bar with a length of $L = 4$ mm and a width of $w = 3.6$ mm by optical lithography and ion beam etching before transport measurements.

**Measurements of static magnetization**. Static magnetization of the sample was investigated in the temperature range of 10–300 K in magnetic fields up to 60 kOe using a conventional Quantum Design Magnetic Property Measurement System SQUID magnetometer. The magnetic properties of the substrate were measured separately, and its contribution was subtracted from the total magnetic moment of the sample.

**Spin pumping measurements**. The sample was mounted on a coplanar waveguide CPW (separated from CPW by an insulating layer) affixed at one end of a custom variable temperature insert and inserted into a Cryogenic Ltd. Cryogen-free Vector Magnet with a 20 kOe rotating field in any direction. The microwave radio frequency field lies in the film plane. An external magnetic field $H$ of the azimuthal angle $\theta_H$ was applied in the film plane. Spin pumping measurements were performed by recording the dc voltage along the Pt bar using a lock-in technique in the frequency range of 1–18 GHz at temperatures of 10–300 K. The microwave power $P_{app}$ was monitored by a microwave power sensor (R&S NRP50T) and a vector network analyser (R&S ZVA24).

## Data availability

Data that support the findings of this study are deposited in Zenodo with the identifier https://doi.org/10.5281/zenodo.5889670.

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

## Acknowledgements

The authors thank Ran Cheng and J. X. Deng for valuable discussions and assistance with the data analysis. This work is supported by the National Key Research and Development Program of China (No. 2017YFA0303303), the National Natural Science Foundation of China (Nos. 11874072 and 11874416), the Strategic Priority Research Program of the Chinese Academy of Sciences (Grant No. XDB33000000), and the National Key Basic Research Project of China (Grant No. 2016YFA0300600).

## Author contributions

Y.L. conducted the static and dynamic measurements. Y.L. and Z.X. built the setup for spin pumping. L.L. performed the micromagnetic simulation. K.Z. helped with the sample growth. Y.S. and P.G. performed EELS and STEM measurements. Y.M. grew samples and contributed to the writing of the manuscript. H.-W.Z. was involved in the discussion. Q.N. proposed the theory and contributed to the writing of the manuscript. J.L. designed the experiments, analyzed the data and wrote the paper.

## Competing interests

The authors declare no competing interests.
