## [Peer Review File · Nature Communications]

REVIEWER COMMENTS

Reviewer #1 (Remarks to the Author):

The authors Liu et. al. have reported their work on the spin pumping in an artificial AFM material. The artificial AFM material is consisted of a multi-stack of Py/Gd (the sample in the paper has three stacks), in which Py and Gd are coupled antiferromagnetically. The authors show that, using a detector (a Pt layer) grown next to the topmost Py layer, the spin current associated to the precession of the magnetization of the Py layer can be determined. The precession can be right-handed or left-handed (opposite on chirality) depending when the sample is in the Py-aligned or the Gd-aligned phases, which is dominated at different temperatures. The authors claim the results demonstrated a way to manipulate the chirality of the AFM magnons and has the potential for AFM spintronics. Yes, the results reported is interesting. It is indeed hard to probe the spin current generated by one spin sub-lattice in a AFM material and control the chirality. The paper shows a potential to probe this using artificial multilayer material. However, based on the current data of the manuscript, I doubt what has been reported is strictly an effect of AFM magnons. In the following I list my questions. I can not recommend the paper based on its current form.

In the sample, the Pt detector detects the spin current generated by the precession of the magnetization of the neighboring Py layer. The authors mentioned that only the Py layer at the interface with the Pt detector matters. The anti-alignment of magnetization between Py and Gd can already be established with one Gd/Py unit. My question is that why the structure needs multiple (three stacks) Py/Gd layers. It looks that the effect only needs the AFM coupling between one Gd layer and one Py layer, and it does not have to be a multi-layer representing an artificial AFM. Can the authors provide further information regarding what is the minimum number of stacks one needs to have in order to demonstrate the reported effect? On the other hand, it is noticed that the FMR frequency is on the order of ten GHz. This brings in the question that if this is an effect of real AFM magnons, which often has much higher frequency. Could it be just an effect of the FM magnon of the Py layer, in which the magnetization of Py is coupled antiferromagnetically to the neighboring magnetic layer (Gd layer)? This also relates to the previous question: it looks that the effect does not need multi-stack of Gd/Py (one stack of Gd/Py that is antiferromagnetically coupled may also work), which makes me doubt if the spin pumping signal is strictly an effect of AFM magnon.

Further, regarding the compensation temperature T_M (Fig 2) and its signatures on the AHE signal (SI Note 3), it is not very clear. As I know, AHE is sensitive to the out-of-plane magnetization. The magnetization M vs. T that is used to determine T_M is in-plane. How can AHE signifies the compensation of an in-plane magnetization? Also, how does the author prove that the Py and Gd layers are antiferromagnetically coupled? Further, the data suggesting H_{twist} (Fig 2) shows a rather small effect (basically only a curvature change on M vs H), while at $H < H_{\text{twist}}$ the data also shows a magnetization increasing with H under a comparable slope. Can this be simply an effect of the alignment of the magnetic domain? Does the author have more evidence pinpointing the twisted state? Are there any signatures from AHE that can be associated to the twisted state?

It would be good to also provide information on the quality of the multi-layer sample to certain extent. It is unclear to me that, while both the Curie temperatures of Py (872 K) and Gd (293 K) are high, why the compensation temperature T_M is so low $\sim 60\text{K}$? How good is the quality of the layers and what are the Curie temperatures of thin film Py and Gd that the authors grown?

Reviewer #2 (Remarks to the Author):

This paper reports on the experimental demonstration of spin pumping from an antiferromagnetically coupled metallic multilayer. The focus is on the magnon chirality as an independent degree of freedom from the net magnetization direction, whereby the chirality-based computation could be constructed in the future. The topic is timely in line with a recent growing interest in antiferromagnetic spintronics that has potential for more functionalities and higher performance overcoming the obstacles faced by standard spintronics as well as conventional electronics. The present title and abstract give rise to great expectations. Soon after that, however, readers might have some disappointment with the rather overselling conclusion. Although the data shown in the paper and analysis are good and convincing the presentation and the positioning of the results should be totally improved. The present manuscript is thus not suitable to warrant publication in Nature Communications. Suggestions and points to be addressed are given below.

In the abstract, the authors claim that their multilayer device represents “an innovatively designed artificial antiferromagnet by which the switching, reading, and modulation of magnon chirality are unambiguously demonstrated.” This sentence contains several levels of misleading statements. And the same goes for the overall tone of the manuscript, which had better be adjusted properly.

First, technically the magnetic-ordered state is not an antiferromagnet but a ferrimagnet with an uncompensated moment. This difference is crucial to understanding the physics behind the observation since the twisted state is specific to the ferrimagnetic order. This point should be distinguished throughout the manuscript for the sake of scientific correctness.

Second, within a specialized field of spintronics, such metallic multilayers capped with the spin-charge conversion Pt layer are quite common platforms for investigation of spin current physics and it is rather embarrassing to regard them as innovative. See for example, K. Tanaka et al., Appl. Phys. Express 7 063010 (2014) where the device structure is similar although the purposes of the study are different.

Third, at a more engineering level, it is hard to consider the alternation of chirality states by temperatures or magnetic field as a meaningful “switching” because the ranges of temperature from 300 K to 10 K and the field of the order of kOe are significantly incompatible of any requirements for the device applications such as energy consumption, switching time scale, and device architecture. All of them are already highly achieved by existing technologies and so what is the advanced point of the present system?

It is inaccurate to account for the different magnetic layers (Py or Gd) as a sublattice. The latter concept was invented to describe the genuine antiferromagnetic and ferrimagnetic orders on the atomic scale. While comparison of the multilayered synthetic systems with genuine ferri/antiferromagnetic systems is sometimes helpful in understanding the underlying physics, stating that “we can prove the spin pumping voltage $V(H)$ of the specific magnetic sublattice” (page 5, line 143) is very misleading.

The present study is more relevant to the compensation phenomena of ferrimagnetic materials that are widely studied in recent years. Especially for the dynamical aspects including the spin pumping effect, the angular momentum compensation is more essential than the magnetization compensation defined by T_M in Fig.2b. For example, the thermal spin pumping [S. Geprgs et al., Nat. Commun. 7, 10452 (2016)], enhancement of domain wall mobility [K.J. Kim et al., Nat. Mater. 16, 1187 (2017)], and the gyromagnetic reversal [M. Imai et al., Appl. Phys. Lett. 113, 052402 (2018)] have been investigated. It is very interesting to know the observation of chirality change in terms of angular momentum

compensation between Py and Gd whose gyromagnetic ratios are known.

The important finding of this work is displayed in Fig.3 h and i, showing that the spin polarization of pumped spin current depends on chirality and not on the equilibrium magnetization direction. While the full angle dependence of the data is located in Note 4 of Supplementary Information but why are they not shown in Fig.3 of the main text (only one data point for each plot)?

The discussion part starting from page 7 exaggerates the merits of the so-called chirality-based computing and logical operation. Unfortunately, this part totally does not make sense when one imagines how such memory bits or logic elements work in the integrated circuits. It is hardly feasible, even at least "in principle", to access each bit, to write different bit states, and to store the bit state non-volatile way. Such overwhelming exaggeration is unnecessary and should be avoided.

Please define/explain "the spin rectification effect" on page 5, line 146, 151, and page 6, line 166.

The microwave power " P_{app} " appears on page 7, line 210 (Fig.4 caption) before the definition in Methods page 9, line 300.

Please correct the inequality equation on page 8, line 259, which is incorrect for vector quantities.

In Supplementary Information Note 1, where \rightarrow which ?

Reviewer #3 (Remarks to the Author):

Liu et al. report on the precessional magnetization dynamics of artificial antiferromagnets detected by high frequency electrical measurements. They study [Gd/Py] multilayers with in-plane magnetization that display robust antiferromagnetic interfacial exchange coupling and a compensation temperature at which the temperature dependent Py and Gd magnetic moments are equal and opposite. This system has been chosen advisedly to allow the authors to study the different ferromagnetic resonance and exchange modes of the stack. Furthermore, a Pt layer was placed on top of the final Py layer so that the precessing Py magnetization pumps spins into the Pt and generates a measurable voltage by means of the inverse spin Hall effect (ISHE). The principal difficulty with making measurements of this kind on metallic stacks is the presence of parasitic effects. Precession is induced by a RF magnetic field generated by a RF current in an underlying coplanar waveguide. This field may also induce eddy currents within the metallic stack that lead to additional voltages via the spin rectification (aka diode) effect (SRE). Separation of contributions from the ISHE and SRE is a painstaking process that requires additional measurements to be made as the orientation of the static applied magnetic field is varied. To their credit, the authors provide supplementary material that includes extensive discussion of these tests and also comparison with other reference samples from which it is inferred that the ISHE contribution dominates in the sample presented in the main text. The authors are able to show that the sign of the measured DC signal depends upon the sense of precession of the last Py layer. By choosing the polarity of the applied static field and its magnitude, they are able to choose both the polarity and sense of precession of the magnetization of the Py magnetization, giving 4 different states (summarised in Figure 5) that might be used to encode information.

This is an impressive piece of work that extends the scope of high frequency spintronics

and so I am happy to recommend publication in Nature Communications. I have a few critical comments that the authors should address prior to publication.

1. I think many readers may be confused by the definition of chirality in this paper. Chirality requires the identification of 3 vectors that form either a right or a left-handed set. When we talk about left and right polarised electromagnetic waves, the wavevector is one of the three vectors, but here the modes are stationary. What are the 3 vectors in the present case?

2. In Figure 1(a),(b) it is not immediately obvious that the precessing vector represents that magnetization of the final Py layer. It would be helpful to explain this within the caption.

3. The discussion of the superposition of ISHE and SRE voltages in the main text assumes a strong degree of familiarity with these effects. While there is more discussion in the supplementary material, it would be useful to the reader to briefly explain the origin of these effects within the main text e.g. the induced current in the stack is not mentioned at all in the main text.

4. At line 259 of page 8/12 I didn't understand the meaning of $\langle m \rangle < +z$. My suspicion is that the "less than" symbol is a font substitution error because there were numerous others in the supplementary information.

5. While the supplementary information provides information about the interdiffusion of the Ni and Gd, there is no other structural information. More information should be provided about the crystallographic structure, texture, and interfacial roughness of the layers. This will certainly be important to other researchers who might attempt to reproduce the results of the present paper.

6. How did the authors level the unsaturated hysteresis loops (e.g. Fig S2(a)) that they show? It is common to perform a linear background subtraction, even after the response of a bare substrate has been subtracted, but this is often difficult if the available field is insufficient for saturation.

To Reviewer #1

We thank Reviewer #1 for his/her appreciation of the significance and novelty of our work. In the following, we list the responses to his/her comments and revisions following his/her suggestions.

The anti-alignment of magnetization between Py and Gd can already be established with one Gd/Py unit. My question is that why the structure needs multiple (three stacks) Py/Gd layers. It looks that the effect only needs the AFM coupling between one Gd layer and one Py layer, and it does not have to be a multi-layer representing an artificial AFM. Can the authors provide further information regarding what is the minimum number of stacks one needs to have in order to demonstrate the reported effect?

We thank Reviewer #1 for giving us this opportunity to clarify the design of the Py/Gd multilayer. Actually, the design of the Py/Gd multilayer is quite tricky. It is challenging to preserve the ferromagnetic order in ultrathin Gd layer. The Curie temperature of bulk Gd is $T_c \sim 293$ K, but it dramatically decreases for ultrathin Gd films due to the size effect. To realize the magnetic Py/Gd layer with the antiferromagnetic coupling, there are two approaches. The first approach is to increase the Gd layer thickness. However, given the Gd magnetostatic exchange length ($\sqrt{A_{ex}/M_s}$) of $2 \sim 3$ nm (J. Robinson et al., Sci. Rep. 2, 699 (2012)), the Gd layer thickness should remain thin enough to preserve the magnetic single-domain state. The magnetic multi-domain may be observed in thick Gd films. The second approach is to increase the repetition number of Py/Gd layer. We present the temperature dependence of in-plane magnetization of Py(2.5)/Gd(3) bilayer, Py(2.5)/Gd(3)/Py(2.5) trilayer, Py(2.5)/Gd(3)/Py(2.5)/Gd(3)/Py(2.5) sample in the following figure (thickness in nm). According to these results, the minimum number of stacks is Py/Gd/Py trilayer. The Py/Gd/Py/Gd/Py multilayer shows more well-defined compensation magnetization with respect to Py/Gd/Py trilayer. And Gd-aligned phase is not accessible in Py/Gd bilayer.

We add this figure and discussion into the supplementary information.

On the other hand, it is noticed that the FMR frequency is on the order of ten GHz. This brings in the question that if this is an effect of real AFM magnons, which often has much higher frequency. Could it be just an effect of the FM magnon of the Py layer, in which the magnetization of Py is coupled antiferromagnetically to the neighboring magnetic layer (Gd layer)? This also relates to the previous question: it looks that the effect does not need multi-stack of Gd/Py (one stack of

Gd/Py that is antiferromagnetically coupled may also work), which makes me doubt if the spin pumping signal is strictly an effect of AFM magnon.

The spin pumping signals are attributed to two different resonance modes of the Py/Gd multilayer. The first one is observed in the collinear coupling regime which is of the ferromagnetic character (referred to as FMR mode). The second one is observed in the twisted state. The dynamic properties of the second mode are governed by the antiferromagnetic coupling between \mathbf{M}_{Py} and \mathbf{M}_{Gd} . Namely, the second resonance mode is of the antiferromagnetic character (referred to as exchange mode) and has been explicitly evidenced in the experiment and micromagnetic simulation. Our experimental and simulation results are in good agreement with the literatures' results (A. Drovosekov et al., J. Magn. Magn. Mater. 475, 668 (2019), J. Phys.: Condens. Matter 29 115802 (2017)).

According to our micromagnetic simulation and the literatures' results (R. Ranchal et al., Phys. Rev. B 85, 024403 (2012), P. Lapa et al., Phys. Rev. B 96, 024418 (2017)), the effective magnetic field H_{AF} of the antiferromagnetic coupling at the Py/Gd interface is \sim few Tesla ($H_{\text{AF}} = J_{\text{Py/Gd}}/M_s t$, here t is the thickness of Py or Gd layer), which is weaker than the nearest-neighbor exchange interaction in natural antiferromagnets (\sim hundred Tesla). Therefore, the resonance frequency of the second mode in the Py/Gd multilayer is on the order of ten GHz, while the resonance modes in natural antiferromagnets are on the order of THz.

It is worth noting that the sublayer thickness of the Py/Gd multilayer is very thin ($t_{\text{Py}} = 2.5$ nm, $t_{\text{Gd}} = 3$ nm). H_{AF} is one order of magnitude stronger than the external field at resonance. Therefore, the antiferromagnetic coupling field H_{AF} dominates the magnetic dynamics of the Py/Gd multilayer, resulting in two dynamic eigenmodes in this system, i.e. FMR mode and exchange mode (Y. Nambu et al., PRL 125, 027201 (2020), S. Geschwind and L. R. Walker, J. Appl. Phys. 30, S163 (1959)). The FM magnon of the Py layer is not the eigenmode and cannot exist in such system.

In summary, the spin pumping signals are attributed to two distinct resonance modes of the Py/Gd multilayer (FMR mode and exchange mode). The second mode is of the antiferromagnetic character.

Further, regarding the compensation temperature T_{M} (Fig 2) and its signatures on the AHE signal (SI Note 3), it is not very clear. As I know, AHE is sensitive to the out-of-plane magnetization. The magnetization \mathbf{M} vs. T that is used to determine T_{M} is in-plane. How can AHE signifies the compensation of an in-plane magnetization? Also, how does the author prove that the Py and Gd layers are antiferromagnetically coupled?

Indeed, AHE is sensitive to the out-of-plane (OOP) magnetization. In the AHE measurement, OOP magnetic field is applied to drive the reorientations of \mathbf{M}_{Py} and \mathbf{M}_{Gd} from in-plane to OOP.

As sketched below, \mathbf{M}_{Py} tends to be parallel to OOP field in Py-aligned phase and antiparallel to OOP field in Gd-aligned phase. The twisting between \mathbf{M}_{Py} and \mathbf{M}_{Gd} may be observed at a sufficiently strong OOP field. It is well-known that AHE of rare-earth-transition-metal ferrimagnet is governed by the transition metal (R. Mishra et al., PRL 118, 167201 (2017)), i.e., \mathbf{M}_{Py} dominates AHE signal in the Py/Gd multilayer. The opposite \mathbf{M}_{Py} orientation leads to the opposite sign of AHE. Therefore, the AHE signals of the opposite sign are observed in Py-aligned and Gd-aligned phases, respectively. In another word, AHE signals change the sign at the compensation temperature T_{M} , as shown in AHE data below.

So far, AHE becomes a conventional method to demonstrate the antiferromagnetic coupling in compensation magnetization system (C. Kim et al., Nat. Mater. 19, 980–985 (2020)). The reversal of AHE sign is a direct evidence for the antiferromagnetic coupling between M_{Py} and M_{Gd} .

In addition, as shown in the answer to the first question, the M-T curve with a compensation temperature T_M can also confirm the antiferromagnetic coupling between M_{Py} and M_{Gd} . Meanwhile, the second resonance mode in the twisted state is an evidence as well.

Further, the data suggesting H_{twist} (Fig 2) shows a rather small effect (basically only a curvature change on M vs H), while at $H < H_{\text{twist}}$ the data also shows a magnetization increasing with H under a comparable slope. Can this be simply an effect of the alignment of the magnetic domain? Does the author have more evidence pinpointing the twisted state? Are there any signatures from AHE that can be associated to the twisted state?

We thank Reviewer for this constructive comment. Generally speaking, the slope of M-H curve at $H < H_{\text{twist}}$ is due to the inhomogeneous magnetization at the Py/Gd interface. This effect has been carefully investigated in Ref. 21 in our manuscript. As shown in supplementary Note 1, such interfacial inhomogeneity is evidenced by EDS mapping (interfacial intermixing).

As suggested by Reviewer, the twisting between M_{Py} and M_{Gd} at a sufficiently strong OOP field can be revealed via AHE measurement. The data below exhibits the $R_{\text{AHE}}-H$ curve at $T = 40$ K. The magnetization configuration is dominated by M_{Gd} when $H < H_{\text{twist}}$. The right figure is the magnified view of the region near $H = 0$ Oe in the left figure (red dashed rectangle). Gd-aligned phase is retained at $H < 1$ kOe and the twisted state begins to develop above 1 kOe. The antiferromagnetic alignment of M_{Py} and M_{Gd} is suppressed by increasing H and the parallel alignment of M_{Py} and M_{Gd} is achieved at $H > 65$ kOe. It's worth noting that the maximum $|R_{\text{AHE}}|$ of Gd-aligned phase is greater than the saturated $|R_{\text{AHE}}|$ of the parallel alignment of M_{Py} and M_{Gd} , indicating that a portion of M_{Py} is not fully reversed by H due to the strong AFM coupling at Py/Gd interface.

AHE is sensitive to OOP magnetization. The additional tools are demanded for probing the in-plane magnetization. As we presented in the manuscript, M-T curve and M-H curve are the conventional method to probe the in-plane magnetization configuration of the compensation magnetization system. In particular, the magnetization configuration of this system has been well studied before. The relevant results were summarized in the review article (*J. Phys.: Condens. Matter* **5**, 3727 (1993)). M-H curve and the additional calculation are widely applied to the detection of the twisted state in this system (*Phys. Rev. B* **96**, 024418 (2017)).

In addition, the in-plane magnetization configuration of this system can also be determined by anisotropic magnetoresistance (AMR) (*Phys. Rev. Materials* **2**, 094404 (2018)). The following figure shows the AMR of the Py/Gd multilayer at $T = 10$ K. $(\Delta R/R)_{\parallel}$ is the MR data for external field $H \parallel$ electric current I . And $(\Delta R/R)_{\perp}$ is the MR data for $H \perp I$. Hence $(\Delta R/R)_{\parallel} - (\Delta R/R)_{\perp}$ denotes the AMR of this sample. The magnitude of AMR remains unchanged at $H < H_{\text{twist}}$. In contrast, the twisting between M_{Py} and M_{Gd} at $H > H_{\text{twist}}$ can induce a dramatic change in AMR due to spin scattering. The AMR in the twisted state is in excellent agreement with the calculation result

based on LZ model (P. Levy and Shufeng Zhang, PRL 79, 5110 (1997)). The systematic study of the AMR in the twisted state will be published elsewhere very soon.

In summary, the twisted state of the Py/Gd multilayer is unambiguously demonstrated by measuring M-H curve and AMR, respectively. The OOP twisting between M_{Py} and M_{Gd} is clearly revealed via AHE.

Additionally, the AMR curve in the twisted state is also a direct evidence for the antiferromagnetic coupling between M_{Py} and M_{Gd} .

It would be good to also provide information on the quality of the multi-layer sample to certain extent. It is unclear to me that, while both the Curie temperatures of Py (872 K) and Gd (293 K) are high, why the compensation temperature T_M is so low $\sim 60\text{K}$? How good is the quality of the layers and what are the Curie temperatures of thin film Py and Gd that the authors grown?

We thank Reviewer for this valuable suggestion. The quality of the Py/Gd multilayer sample has been characterized by energy-dispersive X-ray spectroscopy (EDS), as shown in Supplementary Note 1. The well-defined interfaces of Py/Gd multilayer were observed. The Py layer and Gd layer are confirmed to be continuous and uniform.

Meanwhile, the quality of the sample is also characterized by the low-angle x-ray reflectivity (XRR) and the high-angle x-ray diffraction (XRD). As shown above, the periodical oscillations observed in the XRR scans confirm the well-defined interfaces of Py/Gd multilayer, in consistent with the

result of EDS mapping. The high-angle XRD scan shows that the Py/Gd multilayer sample is polycrystalline. Different textures were observed for Gd layer and Pt capping layer. We add the XRR and XRD data into the Supplementary Note 1.

The Curie temperature is $T_C^{Py} = 872$ K for bulk Py and $T_C^{Gd} = 293$ K for bulk Gd. As mentioned in the answer to the first question, T_C dramatically decrease for ultrathin films due to the size effect. In fact, the temperature dependent magnetization of such multilayer system has been very well studied. The strong antiferromagnetic coupling at interface significantly enhances T_C^{Gd} of Gd layer (D. Haskel et al., PRL 87, 207201 (2001)). Consequently, T_C^{Gd} of Gd layer becomes very close to T_C^{Py} of Py layer (P. Lapa et al., PRB 96, 024418 (2017)). As shown in the schematic drawing below, then the different curvatures of M_{Py} -T curve and M_{Gd} -T curve lead to the compensation temperature T_M which is lower than T_C^{Gd} and T_C^{Py} . This T_M is tunable by changing the sublayer thickness or the repetition number of the multilayer, as presented in the answer to the first question. In our case, $T_M = 60$ K is a normal value of compensation temperature in Py/Gd multilayer system.

This figure sketches M-T curves of Py layer and Gd layer in Py/Gd multilayer. T_C^{Gd} of Gd layer is very close to T_C^{Py} of Py layer. The different curvatures of M_{Py} -T curve and M_{Gd} -T curve lead to the compensation temperature $T_M < T_C$.

To Reviewer #2

We thank Reviewer #2 for his/her appreciation of our work, as “the data shown in the paper and analysis are good and convincing”. We are very grateful for his/her suggestions to improve the presentation and the positioning of the results. In particular, we emphasize the result in Fig. 3 in the abstract following Reviewer #2’s suggestions.

In the following, we outline the revisions following his/her suggestions and the responses to his/her comments.

First, technically the magnetic-ordered state is not an antiferromagnet but a ferrimagnet with an uncompensated moment. This difference is crucial to understanding the physics behind the observation since the twisted state is specific to the ferrimagnetic order. This point should be distinguished throughout the manuscript for the sake of scientific correctness.

We thank Reviewer #2 for this critical suggestion. The term “innovatively designed artificial antiferromagnet” is revised to be “artificial ferrimagnet” in the manuscript thoroughly. The adjective “innovatively designed” is removed from the manuscript.

Second, within a specialized field of spintronics, such metallic multilayers capped with the spin-charge conversion Pt layer are quite common platforms for investigation of spin current physics and it is rather embarrassing to regard them as innovative. See for example, K. Tanaka et al., Appl. Phys. Express 7 063010 (2014) where the device structure is similar although the purposes of the study are different.

We thank Reviewer #2 for this comment. As mentioned above, the adjective “innovatively designed” is removed from the manuscript thoroughly.

Third, at a more engineering level, it is hard to consider the alternation of chirality states by temperatures or magnetic field as a meaningful “switching” because the ranges of temperature from 300 K to 10 K and the field of the order of kOe are significantly incompatible of any requirements for the device applications such as energy consumption, switching time scale, and device architecture. All of them are already highly achieved by existing technologies and so what is the advanced point of the present system?

We thank Reviewer #2 for this constructive question. At present, the operations of magnonic devices are based on the amplitude and phase. However, the chirality of magnons is an intrinsic degree of freedom, thus should be more robust in the operations. In particular, the magnetic states of high-dimensionality may become possible in the linear combinations of the magnon chiralities (M. Daniels et al., PRB 98, 134450 (2018)). Therefore, chirality-based spintronics is a prospective research field.

To the best of our knowledge, this work is the first demonstration of magnon chirality as an independent degree of freedom. It is the first time that magnon chirality can be readily switched when preserving the magnetization equilibrium direction. In particular, it is unambiguously demonstrated that the polarity of spin pumping is determined by the chirality of spin precession.

As a pioneering work in chirality-based spintronics, the “switching of magnon chirality” is not efficient enough (an external field of the order of kOe is required). We totally agree with Reviewer #2 on this comment. From the application point of view, great efforts are demanded for the optimization of this “switching”. Thanks to the Py/Gd multilayer system, the optimization of this

“switching” is possible because the switching field ΔH can be decreased by decreasing H_{twist} (Fig. 4(d)). Hence, it is a promising and exciting topic for future research on how to optimize this “switching” of magnon chirality.

Finally, to address the Reviewer’s concern, we put less emphasis on the discussion of the applications in the manuscript. The discussions about “computing”, “logical operations”, “chiral-magnon devices”, etc., are removed from the manuscript.

It is inaccurate to account for the different magnetic layers (Py or Gd) as a sublattice. The latter concept was invented to describe the genuine antiferromagnetic and ferrimagnetic orders on the atomic scale. While comparison of the multilayered synthetic systems with genuine ferri/antiferromagnetic systems is sometimes helpful in understanding the underlying physics, stating that “we can prove the spin pumping voltage $V(H)$ of the specific magnetic sublattice” (page 5, line 143) is very misleading.

We thank Reviewer #2 for this constructive comment. “magnetic sublattice” is revised to be “magnetic sublayer” in the manuscript thoroughly.

The present study is more relevant to the compensation phenomena of ferrimagnetic materials that are widely studied in recent years. Especially for the dynamical aspects including the spin pumping effect, the angular momentum compensation is more essential than the magnetization compensation defined by T_M in Fig.2b. For example, the thermal spin pumping [S. Geprgs et al., Nat. Commun. 7, 10452 (2016)], enhancement of domain wall mobility [K.J. Kim et al., Nat. Mater. 16, 1187 (2017)], and the gyromagnetic reversal [M. Imai et al., Appl. Phys. Lett. 113, 052402 (2018)] have been investigated. It is very interesting to know the observation of chirality change in terms of angular momentum compensation between Py and Gd whose gyromagnetic ratios are known.

We thank Reviewer #2 for this constructive suggestion. Indeed, the observation of chirality switching and chirality-dependent spin pumping in such momentum compensation system is a very interesting topic. We believe this topic deserves a great effort and would receive intense interest in the future. And we are very glad to cite the above-mentioned literatures as the references in the introduction of manuscript.

The important finding of this work is displayed in Fig.3 h and i, showing that the spin polarization of pumped spin current depends on chirality and not on the equilibrium magnetization direction. While the full angle dependence of the data is located in Note 4 of Supplementary Information but why are they not shown in Fig.3 of the main text (only one data point for each plot)?

We thank Reviewer #2 for this constructive suggestion. The angular dependent data in Fig. S4 is added into Fig. 3 for a better visibility. As the important finding of this work, we emphasize the result of Fig. 3 in the abstract.

The discussion part starting from page 7 exaggerates the merits of the so-called chirality-based computing and logical operation. Unfortunately, this part totally does not make sense when one imagines how such memory bits or logic elements work in the integrated circuits. It is hardly feasible, even at least “in principle”, to access each bit, to write different bit states, and to store the bit state non-volatile way. Such overwhelming exaggeration is unnecessary and should be avoided.

We thank Reviewer #2 for this constructive suggestion. The discussion part of “chirality-based computing and logical operation” is completely removed from the manuscript.

Please define/explain “the spin rectification effect” on page 5, line 146, 151, and page 6, line 166.

The microwave field could induce an ac current inside the FM metal. Meanwhile, the oscillating magnetization of the FM metal can induce a time dependent resistance by AMR, AHE, etc. The interplay between the ac current and time dependent resistance may induce the spin rectification effect, leading to the DC voltage signals of symmetric and antisymmetric Lorentzians. We add this definition into the manuscript.

The microwave power “ P_{app} ” appears on page 7, line 210 (Fig.4 caption) before the definition in Methods page 9, line 300.

We add “microwave power P_{app} ” in front of Fig. 4.

Please correct the inequality equation on page 8, line 259, which is incorrect for vector quantities.

In Fig. 5 caption, the sentence should be “This degree of freedom is denoted by $\langle \uparrow \rangle$ for $\langle \mathbf{m} \rangle // +\mathbf{z}$ or $\langle \mathbf{M} \rangle // +\mathbf{z}$, and by $\langle \downarrow \rangle$ for $\langle \mathbf{m} \rangle // -\mathbf{z}$ or $\langle \mathbf{M} \rangle // -\mathbf{z}$ ”. This typo is a font substitution error.

In Supplementary Information Note 1, where \rightarrow which ?

We change “where” to “which” in supplementary Note 1.

To Reviewer #3

We thank Reviewer #3 for his/her appreciation of our work, as “This is an impressive piece of work that extends the scope of high frequency spintronics”. In the following, we outline the revisions following his/her valuable comments.

1. I think many readers may be confused by the definition of chirality in this paper. Chirality requires the identification of 3 vectors that form either a right or a left-handed set. When we talk about left and right polarised electromagnetic waves, the wavevector is one of the three vectors, but here the modes are stationary. What are the 3 vectors in the present case?

We thank Reviewer #3 for this critical reminder. The chirality of magnon is defined by the precessing magnetic moment \mathbf{M} and the magnetization equilibrium direction (employed as the vector). Taking Fig. 1 as an example (shown in the following figure), chirality is identified as right-handed (left-handed) for the counter-clockwise (clockwise) \mathbf{M} precession around the equilibrium direction. Our spin pumping measurements were performed in magnon mode with wavevector $\mathbf{k} = 0$. The definition of chirality is still valid when $\mathbf{k} \neq 0$. We add this definition into Fig. 1 caption.

2. In Figure 1(a),(b) it is not immediately obvious that the precessing vector represents that magnetization of the final Py layer. It would be helpful to explain this within the caption.

We thank Reviewer #3 for this suggestion. The instruction of the “outermost Py layer” is added into Fig. 1 caption.

3. The discussion of the superposition of ISHE and SRE voltages in the main text assumes a strong degree of familiarity with these effects. While there is more discussion in the supplementary material, it would be useful to the reader to briefly explain the origin of these effects within the main text e.g. the induced current in the stack is not mentioned at all in the main text.

We thank Reviewer #3 for this constructive suggestion. The microwave field could induce an ac current inside the FM metal. Meanwhile, the oscillating magnetization of the FM metal can induce a time dependent resistance by AMR, AHE, etc. The interplay between the ac current and time

dependent resistance may induce the spin rectification effect, leading to the DC voltage signals of symmetric and antisymmetric Lorentzians. We add this description into the main text.

4. At line 259 of page 8/12 I didn't understand the meaning of $\langle m \rangle < +z$. My suspicion is that the "less than" symbol is a font substitution error because there were numerous others in the supplementary information.

We thank Reviewer #3 for this correction. Indeed, this typo is a font substitution error. This sentence should be "This degree of freedom is denoted by $\langle \uparrow \rangle$ for $\langle m \rangle // +z$ or $\langle M \rangle // +z$, and by $\langle \downarrow \rangle$ for $\langle m \rangle // -z$ or $\langle M \rangle // -z$ ".

5. While the supplementary information provides information about the interdiffusion of the Ni and Gd, there is no other structural information. More information should be provided about the crystallographic structure, texture, and interfacial roughness of the layers. This will certainly be important to other researchers who might attempt to reproduce the results of the present paper.

We thank Reviewer #3 for this constructive suggestion. The quality of the sample is also characterized by the low-angle x-ray reflectivity (XRR) and the high-angle x-ray diffraction (XRD). As shown in the figure below, the periodical oscillations observed in the XRR scans confirm the well-defined interfaces of Py/Gd multilayer, in consistent with the result of EDS mapping. The fitting of XRR reveals a smooth Py/Gd interface with an RMS roughness of 0.49 nm. The high-angle XRD scan shows that the Py/Gd multilayer sample is polycrystalline. Different textures were observed for Gd layer and Pt capping layer. We add the XRR and XRD data into the Supplementary Note 1.

6. How did the authors level the unsaturated hysteresis loops (e.g. Fig S2(a)) that they show? It is common to perform a linear background subtraction, even after the response of a bare substrate has been subtracted, but this is often difficult if the available field is insufficient for saturation.

Actually, the saturation is achieved when recording all the hysteresis loops (H is up to 6T). As shown below, the data points of 5T-6T are used as the reference. The following is the hysteresis loop of the Py/Gd multilayer at $T = 10$ K. The recorded magnetization has reached saturation at $H < 5$ T.

REVIEWERS' COMMENTS

Reviewer #1 (Remarks to the Author):

The authors Liu et. al. have partially addressed my concerns in their report and in the revised manuscript. I have no doubt on the experiment itself, and the results are clear. However, I do feel that the article is overselling the idea of AFM magnons according to their material system. Therefore, I don't think the manuscript is significant enough to support chirality-based spintronics.

The key issue of the manuscript is the material. I saw that the authors have adopted the new notation as "artificial ferrimagnet (FiMs)" instead of "artificial antiferromagnet (AFM)". Strictly speaking, the multilayer heterostructure used in this work cannot be considered as either of them. Both FiMs and AFM refers to the spin configuration at the sublattice level, or at the atomic scale. However, each layer in the multilayer sample used in this work has a thickness of several nanometers. It is not a true superlattice sample. Thus, it is no accurate to talk about the spin dynamics or magnons of the sublattices, which have opposite chirality. This also agrees with the issue why the FMR frequency is low. As mentioned by the authors in the rebuttal letter, it is because the antiferromagnetic coupling between the thin film layers is much weaker than the exchange coupling between the nearest neighbors at the atomic scale in a true AFM. The FMR signal observed in this work is due to the precession of the magnetization in the Py layer, which is antiferromagnetically coupled to the Gd layer. Actually, only the topmost Py layer matters. It is different from the magnon modes in a AFM, which is a effect of the precession of the whole sublattice. The mechanism behind is the antiferromagnetic coupling between the thin film Py and Gd layers, and it is because of this antiferromagnetic coupling the authors are able to observe the spin pumping signals with opposite signs. However, strictly speaking this is different from the magnons modes in a AFM.

Reviewer #2 (Remarks to the Author):

The authors have satisfactorily addressed the points raised in the reviewers' reports. The presentation and positioning of the results are properly amended and additional information has been provided making the manuscript clearer. Since the scientific idea that leverages chirality as a useful degree of freedom in magnetism is interesting the present work would be an important step to proceed in such a research direction. I, therefore, support the publication of this manuscript in Nature Communications after improving the readability suggested below.

- 1) Vectors must be either bold font or italic font with a right-arrow on them. Please check them throughout the manuscript and figure legends [see Fig.3(b), (c)].
- 2) On page 2, Fig. 1(c)-(e): vector quantities cannot be with inequality. The absolute values (or the saturation magnetization of a scalar quantity) should be used here, such as $|M_A| > |M_B|$.
- 2) On page 3, line 76: the net moment should be the absolute value of the sum of vectors M_A and M_B , i.e., $|M_A + M_B|$.
- 3) On page 3, line 80: vector quantities cannot be with inequality. The absolute values (or the saturation magnetization of a scalar quantity) should be used here, such as $|M_A| > |M_B|$.
- 4) On page 4, line 133: Since $M_{\{Py\}}$ is compared with +z direction is it a vector? If yes, please amend the notation of it and what follows.
- 5) On page 6, line 194: in addition to Ref.20 as a reference to SRE, K. Ando et al, Journal of Applied Physics 109, 103913 (2011) is recommended to be included as a complementary reference to the basics of the phenomenon.

Reviewer #3 (Remarks to the Author):

The authors have responded appropriately to all of my comments and I am impressed by the depth of supporting data upon which they are able to call. My recommendation is therefore to publish.

The other reviewers' comments on Reviewer #1's new report:

"In my eyes, Reviewer #1 seems concerned that the results reported in this work are not genuine (atomic scale) antiferromagnetic (AFM)/ferrimagnetic (FiM) physics. For a proof of concept study, however, it is sometimes helpful and meaningful to model genuine AFM/FiM systems by using synthetic (multilayered) AFM/FiM ones with lower excitation energies. I think that the present study has been carried out in that manner. To clarify the purpose of this study and to avoid the confusion as such happened to Reviewer #1, it can be helpful to advise the authors to insert some sentences explaining that their system is designed to MODEL the FiM physics for proving the chirality based functionality."

We would like you to revise the manuscript according to the above comments, and soften the main claims.

We thank reviewers for the constructive comment. Following the editor's instructions, we insert the clarification of our multilayer design in the second paragraph on page 3, and soften our main claim in the abstract.

To Reviewer #1

We thank Reviewer #1 for his/her comment "I have no doubt on the experiment itself, and the results are clear."

The key issue of the manuscript is the material. I saw that the authors have adopted the new notation as "artificial ferrimagnet (FiMs)" instead of "artificial antiferromagnet (AFM)". Strictly speaking, the multilayer heterostructure used in this work cannot be considered as either of them. Both FiMs and AFM refers to the spin configuration at the sublattice level, or at the atomic scale. However, each layer in the multilayer sample used in this work has a thickness of several nanometers. It is not a true superlattice sample. Thus, it is not accurate to talk about the spin dynamics or magnons of the sublattices, which have opposite chirality. This also agrees with the issue why the FMR frequency is low. As mentioned by the authors in the rebuttal letter, it is because the antiferromagnetic coupling between the thin film layers is much weaker than the exchange coupling between the nearest neighbors at the atomic scale in a true AFM. The FMR signal observed in this work is due to the precession of the magnetization in the Py layer, which is antiferromagnetically coupled to the Gd layer. Actually, only the topmost Py layer matters. It is different from the magnon modes in a AFM, which is an effect of the precession of the whole sublattice. The mechanism behind is the antiferromagnetic coupling between the thin film Py and Gd layers, and it is because of this antiferromagnetic coupling the authors are able to observe the spin pumping signals with opposite signs. However, strictly speaking this is different from the magnons modes in a AFM.

The main purposes of this work are to study chirality-dependent spin pumping and to synthesize a practical platform for chirality-based spintronics. The original proposal of this topic is based on natural AFMs. However, after consuming great efforts on several AFM systems, scientists realized the substantial weakness of AFM systems (see the introduction and Fig. 5 of this manuscript). Nowadays, the research on this topic is stagnating at the first step due to the substantial obstacles

of controlling and detecting magnon chirality. In our work, by synthesizing an artificial ferrimagnet, we successfully demonstrate the switching, reading, and modulation of magnon chirality. All these operations are inaccessible in natural AFMs, suggesting a great advantage of our artificial ferrimagnet over natural AFMs.

Therefore, the material in the manuscript is not an issue but a salient merit. Our artificial ferrimagnet is an ideal platform for chirality-based spintronics, and our work opens the door for the chirality-based information processing and computing.

To Reviewer #2

We thank Reviewer #2 for his/her recommendation of our manuscript. In the following, we response to his/her comments point by point.

1) Vectors must be either bold font or italic font with a right-arrow on them. Please check them throughout the manuscript and figure legends [see Fig.3(b), (c)].

We thank Reviewer #2. The vectors have been formatted to be bold font throughout the manuscript.

2) On page 2, Fig. 1(c)-(e): vector quantities cannot be with inequality. The absolute values (or the saturation magnetization of a scalar quantity) should be used here, such as $|M_A| > |M_B|$.

We thank Reviewer #2. This mistake has been corrected.

2) On page 3, line 76: the net moment should be the absolute value of the sum of vectors M_A and M_B , i.e., $|M_A + M_B|$.

We thank Reviewer #2. This mistake has been revised.

3) On page 3, line 80: vector quantities cannot be with inequality. The absolute values (or the saturation magnetization of a scalar quantity) should be used here, such as $|M_A| > |M_B|$.

We thank Reviewer #2. This mistake is corrected.

4) On page 4, line 133: Since $M_{\{Py\}}$ is compared with +z direction is it a vector? If yes, please amend the notation of it and what follows.

We thank Reviewer #2. $M_{\{Py\}}$ is a vector, and it has been formatted throughout the manuscript.

5) On page 6, line 194: in addition to Ref.20 as a reference to SRE, K. Ando et al, Journal of Applied Physics 109, 103913 (2011) is recommended to be included as a complementary reference to the basics of the phenomenon.

We thank Reviewer #2. This literature has been cited as the reference next to Ref. 20.

To Reviewer #3

We thank Reviewer #3 for his/her recommendation of our manuscript.